# Use of Generalized Additive Model to Detect the Threshold of δ-Aminolevulinic Acid Dehydratase Activity Reduced by Lead Exposure

**DOI:** 10.3390/ijerph17165712

**Published:** 2020-08-07

**Authors:** Chan-Ching Huang, Chen-Cheng Yang, Te-Yu Liu, Chia-Yen Dai, Chao-Ling Wang, Hung-Yi Chuang

**Affiliations:** 1Department of Occupational and Environmental Medicine, Kaohsiung Medical University Hospital, Kaohsiung 807, Taiwan; ymeoung@gmail.com (C.-C.H.); abcmacoto@gmail.com (C.-C.Y.); daichiayen@gmail.com (C.-Y.D.); florawang0913@gmail.com (C.-L.W.); 2Graduate Institute of Medicine, College of Medicine, Kaohsiung Medical University, Kaohsiung 807, Taiwan; 3Department of Public Health, and Research Center for Environmental Medicine, Kaohsiung Medical University, Kaohsiung 807, Taiwan; chy@tpts1.seed.net.tw

**Keywords:** generalized additive model (GAM), blood lead, delta-aminolevulinic dehydratase (ALAD), ALAD polymorphism, hemopoietic enzyme

## Abstract

Background: Lead inhibits the enzymes in heme biosynthesis, mainly reducing δ-aminolevulinic acid dehydratase (ALAD) activity, which could be an available biomarker. The aim of this study was to detect the threshold of δ-aminolevulinic acid dehydratase activity reduced by lead exposure. Methods: We collected data on 121 lead workers and 117 non-exposed workers when annual health examinations were performed. ALAD activity was determined by the standardized method of the European Community. ALAD G177C (rs1800435) genotyping was conducted using the polymerase chain reaction and restricted fragment length polymorphism (PCR-RFLP) method. In order to find a threshold effect, we used generalized additive models (GAMs) and scatter plots with smoothing curves, in addition to multiple regression methods. Results: There were 229 ALAD1-1 homozygotes and 9 ALAD1-2 heterozygotes identified, and no ALAD2-2 homozygotes. Lead workers had significantly lower ALAD activity than non-exposed workers (41.6 ± 22.1 vs. 63.3 ± 14.0 U/L, *p* < 0.001). The results of multiple regressions showed that the blood lead level (BLL) was an important factor inversely associated with ALAD activity. The possible threshold of BLL affecting ALAD activity was around 5 μg/dL. Conclusions: ALAD activity was inhibited by blood lead at a possible threshold of 5 μg/dL, which suggests that ALAD activity could be used as an indicator for lead exposure regulation.

## 1. Introduction

Although leaded gasoline was phased out in Taiwan in 2000, lead continues to be a public health concern due to its industrial uses. Workers exposed to lead have been reported to have blood lead concentrations in the range of 20–50 ug/dL [1]. Chronic lead exposure has been associated with decreasing hemoglobin, which is one of the mechanisms that induces anemia by lead [2]. Lead inhibits three enzymes in the heme biosynthesis pathway, δ-aminolevulinic acid dehydratase (ALAD), coporphyrinogen oxidase, and ferrochelatase, and its effects on ALAD are the most profound. Lead inhibits ALAD stoichiometrically, and degree of erythrocyte inhibition has been used clinically to gauge the degree of lead poisoning [3,4]. ALAD inhibition results in a buildup of aminolevulinic acid detectable in the plasma and urine. At the molecular level, lead displaces a zinc ion at the metal binding site, not the active site, producing inhibition through a change in the quaternary structure of the enzyme [5,6,7]. Thus, ALAD activity may be negatively related to higher blood lead levels.

As a biomarker of lead exposure and availability, the value of measuring ALAD activity has not been well established. An ALAD activity assay could and should be a diagnostic bioassay to evaluate both acute and chronic anemia induced by occupational and environmental exposure to lead and differentiate it from common causes of anemia, such as malnutrition or heredity. The determination of ALAD activity is well suited as a measure of lead-induced anemia to differentiate iron deficiency or inhibition of hemoglobin synthesis. An established threshold should be available for determining ALAD activity.

Methods for detecting a threshold are sophisticated. Traditionally, we would establish a toxicodynamic model and dose-response relationship or curve. However, in 1990, Hastie and Tibshirani developed the generalized additive model [8], a generalized linear model with a linear predictor involving a sum of smooth functions of covariates:g(E(Y)) = b_0_ + f(x_1_) + f(x_2_) + … + f(x_n_)
where f is a smooth function.

This model was helpful for modeling nonlinear variables such as dose–response curves. Since the GAM plot automatically shows nonlinearity with some knots, we could use the knots (inflection or reflected points) to find the threshold effects. The GAM plot used to find a possible threshold helped us determine that lead might cause sensory neuropathy with an effect threshold corresponding to a five year mean blood lead concentration of 31 μg/dL [9].

On the other hand, ALAD genotypes were reported as modifications of the toxic effects of lead. The ALAD G177C (rs1800435) polymorphism (ALAD 1-1, ALAD 1-2, and ALAD 2-2) was mostly studied [10,11,12,13]. However, it was difficult to decide which genotype was in fact the “at-risk” genotype, because previous research used different outcome measures to indicate whether one genotype was more susceptible to adverse effects than the others [14,15]. In addition, few studies determined ALAD activity in the different ALAD genotypes directly, which was another reason why it was difficult to decide which was the “at-risk” genotype. Meanwhile, few studies in occupational and environmental health have determined the ALAD genotype in Taiwan. Hsieh et al. [10] reported ALAD genotypes in the general population, showing that most Taiwanese were ALAD 1-1 (95.4%), and a few were ALAD 2-2. However, the study found that individuals with ALAD 2 alleles had higher blood lead levels than those with ALAD 1 alleles (7.83 ± 5.95 vs. 6.51 ± 5.03 μg/dL). Mani et al. also reported that ALAD 1-2 carriers presented higher blood lead levels and lower ALAD activity than ALAD 1-1 carriers [16]. Studies using genetic markers of susceptibility to environmental and occupational lead toxicity raised the question of whether genes can make certain individuals more vulnerable to environmental and occupational lead. The latter is currently the most interesting question in industrial and environmental health.

The goal of this study was to detect the threshold of δ-aminolevulinic acid dehydratase activity reduced by lead exposure and determine if ALAD activity was modified by ALAD genotypes.

## 2. Materials and Methods

### 2.1. Subjects

The study was carried out by our investigators in a lead factory when annual health examinations were being performed; we asked the workers if they would agree to participate in a study that checked their ALAD genotype and activity in addition to the regular health examination, and we obtained consent forms [17]. For the control group, 117 age- and sex-matched workers from the same area, but different plants, without the use of lead in their industries, were selected to serve as the control group; their socioeconomic status was similar to that of the lead workers. Workers with renal or liver dysfunction or diabetes and pregnant women were excluded. The study protocol was approved by the institutional review board of Kaohsiung Medical University (IRB approval No. KMUH-IRB-950305), and informed consent was obtained from all subjects before the study. Finally, 238 workers agreed to participate in this research and signed the consent forms.

### 2.2. Health Examination

Under the regulations of labor health protection in Taiwan, health examinations should be performed for lead workers annually, including a physical examination, blood lead test, hematologic tests (including hemoglobin, hematocrit, complete blood count), liver function test (alanine aminotransferase (ALT)), and renal function test (including serum creatinine and urine routine test). These examinations and tests were performed by our researchers and occupational physicians; meanwhile, blood and urine samples were frozen for the analysis of genotypes and other tests in the central laboratory of our hospital. Body mass index (BMI) was measured by taking their body weight and height and calculating by weight (kilograms)/height (meters) squared. A short questionnaire on job title, medical and working history, family history, and alcohol and cigarette consumption was administered during the health examination.

### 2.3. DNA Extraction

DNA extraction was managed with the quick start protocol of the QIAamp^®^ Blood Mini Kit as follows:

An approximately 5 mL venous blood sample was drawn from the participant into a tube containing heparin and centrifuged immediately at 3000 rpm for 10 min to separate plasma and serum. Then, 20 μL Qiagen Protease was pipetted into a 1.5 mL microcentrifuge tube and a 200 μL sample was added, followed by 200 μL buffer. This was mixed thoroughly by vortexing, and incubated at 56 °C for 10 min. The 1.5 mL microcentrifuge tube was centrifuged to remove drops from the lid. The mixture was pipetted onto a QIAamp Mini spin column (in a 2 mL collection tube) and centrifuged at 8000 rpm for 1 min. The flow-through and collection tube were discarded, then a new 2 mL collection tube was placed on the QIAamp Mini spin column, and 500 μL AW1 buffer was added. This was centrifuged at full speed (14,000 rpm) for 3 min, then the flow-through and collection tube were discarded. A new 1.5 mL microcentrifuge tube was placed in the QIAamp Mini spin column and 200 μL buffer AE or distilled water was added, and it was incubated at room temperature (15–25 °C) for 1 min. Then, it was centrifuged at 8000 rpm for 1 min to elute the DNA.

### 2.4. ALAD Genotyping

All (*n* = 238) genotyping was conducted by PCR amplification, followed by polymorphism-specific restriction enzyme digestion and gel analysis. ALAD genotyping was conducted by using the polymerase chain reaction and restricted fragment length polymorphism (PCR-RFLP) method. Briefly, the initial amplification used primers 50-AGACAGACATTAGCTCAGTA-30 and 50-GGCAAAGACCACGTCCATTC-30, and thermal cycling was conditioned as initial denaturation at 95 °C for 5 min followed by 35 cycles of denaturation at 95 °C for 30 s, annealing at 56 °C for 30 s, elongation at 72 °C for 30 s, and a final extension of 72 °C for 10 min to generate a 916 bp fragment. The amplified fragment was cleaved by a restriction enzyme, endonuclease MspI [10,18].

### 2.5. ALAD Activity

ALAD activity was measured within 6 h after sample collection using the European standardized method, with spectrophotometric (Unicam Helios UV-VIS spectrophotometer, Thermo Spectronic, Cambridge, UK) determination carried out at 555 nm [19,20]. The principle of the method of determining ALAD activity was based on incubation of the enzyme with excess δ-aminolevulinic acid. ALAD activity was measured by determining the amount of porphobilinogen formed under standard assay conditions. Porphobilinogen that was formed within a fixed time was mixed with modified Ehrlich’s reagent, and the color that developed was measured photometrically against a blank. The quantity of porphobilinogen produced was the measurement of ALAD activity.

The protocol used was adapted from the European standardized method of Berlin and Schaller [19]. The following reagents were prepared in advance: (1) buffers: 0.1 M Na2HPO4 and 0.1 M NaH2PO4; (2) 10 mM ALA in buffer, with the pH adjusted as required for the species being assayed; (3) 0.612 M trichloroacetic acid (TCA); (4) modified Ehrlich’s reagent: 2.5 g p-dimethylaminobenzaldehyde was dissolved in 50 mL of glacial acetic acid in a perchloric acid hood, and 24.5 mL of 70% perchloric acid was added and diluted to 100 mL with glacial acetic acid in a volumetric flask. For quality control, all samples were measured in triplicate and coefficients of variation (CVs) less than 5% were accepted. Enzyme activity was expressed as units per liter (U/L) of erythrocytes (red blood cells (RBCs)) per minute and was calculated according to the following equation:ALAD activity = (Abs × 100 × 2 × DF)/(Hct × 60 × 0.062)
where Abs is the absorbance of the sample, 2 represents the δ-ALAD conversion factor to porphobilinogen, DF is the dilution factor (35 was used in the study), 60 is the incubation time (minutes), 0.062 is the coefficient (in liters per micromole × centimeters), and Hct is hematocrit (percent).

### 2.6. Statistical Analysis

To summarize the data, descriptive statistics were used to calculate the means of continuous variables, including blood lead levels, age, and biochemistry. For category variables such as gender and smoking and alcohol consumption, rates and proportions were used. The comparison of continuous variables between lead-exposed and non-exposed groups used independent t-tests. A chi-square or Fisher’s exact test was applied to compare category variables. Regression analysis was used for the association of ALAD activity and blood lead level with adjustments of other confounders, such as lifestyle factors. In addition, the effects of interactions between lead and ALAD activity or other biochemical parameters were estimated by multiple regressions.

Furthermore, we used generalized additive models (GAMs) and scatter plots with smoothing curves to look at the curve shape of the relationship between ALAD activity and blood lead level. The aim of these techniques was to look for a dose–response curve and identify whether a threshold effect was present [9,21]. The GAMs and scatter plots were set with loess smoothing with 70% span. In addition, the GAMs also contained adjustments of confounders age, gender, BMI, fasting blood sugar, tobacco smoking, and alcohol consumption (more than twice a week). Once a threshold was identified by visual inspection of the fitted curve, categorized blood lead levels to fit the multiple regression analysis were applied to the data. Significance was set at 0.05 (two-tailed). All statistical operations were performed using R, SAS version 9.4, and SPSS version 20.

## 3. Results

The results of the comparison between the lead exposed and non-exposed groups are shown in Table 1. The proportion of men in the exposure group was significantly higher in the non-exposed group (81.0% and 61.1%, respectively; *p* = 0.01), and there was no difference in age, BMI, hemoglobin, and blood glucose levels between the two groups. Average hematocrit (Hct) was slightly higher in the lead-exposed group than the non-exposed group, but no statistical significance was noted (*p* = 0.06). The proportions of smoking and drinking in the lead-exposed group were significantly higher than in the non-exposed group. The mean of blood lead concentration was 19.7 ± 14.7 and 2.9 ± 1.8 ug/dL (mean ± sd, *p* < 0.001) in the exposed and non-exposed groups, and ALAD activity was 41.58 ± 22.07 and 63.32 ± 14.01 U/L (*p* < 0.001) in the two groups, respectively. In addition, the genotypes of ALAD G177C (rs1800435) were not significantly different between the two groups. There were 229 workers (96.2%) with ALAD type 1-1 and 9 workers (3.8%) with ALAD type 1-2, and no one with ALAD 2-2 type was found in our study. There was no significantly different ALAD activity between ALAD G177C (rs1800435) genotypes, ALAD 1-1 and ALAD 1-2.

The number of years on the job (only lead-exposed group) had a significant (stable) negative association with ALAD activity (r = −0.375, *p* < 0.01). However, it was highly correlated with age (r = 0.646, *p* < 0.01). Considering only the lead-exposed group and the variable of number of years on the job and collinearity with age, we used age (all participants) as a covariate in the next step, regression and GAM analysis.

For the 238 workers, multiple linear regression was used to show the association between blood lead level and ALAD enzyme activity. As shown in Table 2, ALAD activity was significantly reduced by 1.04 U/L (*p* < 0.001) for every 1 ug/dL of increased blood lead. The ALAD type (1-2 vs. 1-1) did not make any significant difference. The variables of male gender and drinking alcohol twice per week showed significantly decreased ALAD enzyme activity. Fasting blood glucose significantly inhibited ALAD enzyme activity; for every 1 mg/dL increase in fasting blood glucose, ALAD enzyme activity decreased by 0.11 U/L (*p* < 0.001). Age, BMI, and smoking habits had no significant effect on ALAD enzyme activity.

Because the ALAD G177C (rs1800435) genotype was not significant and only nine persons had the 1-2 type, we focused on the 229 people with ALAD G177C (rs1800435) 1-1 type. The results of the 229 workers with 1-1 type are shown in Table 2, and adjusted R square was more than the model of 238 workers (0.644 vs. 0.639).

The correlation coefficient (R) was −0.760 (R^2^ = 0.58, *p* < 0.001), as shown in Figure 1a. We used generalized additive models (GAMs) and scatter plots with smoothing curves to find an inflection point around 5 ug/dL of the blood lead level (Figure 1b). In addition, the Akaike information criterion (AIC), which has an expected value close to the SSE/n (mean of sum of square error), was used to select the optimal model fitting. Hastie and Tibshirani called AIC the predicted square error that can be used for selection of degrees of freedom or models in hypothesis testing [8]. A small AIC value means a small error in the model fitting. The AIC for multiple regression (Figure 1a) was 38,397.6, and AIC of GAM (Figure 1b) was 36,688.9, which shows that GAM had better prediction ability than the multiple regression model.

To further explore the effects of blood lead concentration on ALAD enzyme activity, 229 workers (excluding ALAD 1-2 type) were divided into nine groups by categorized blood lead concentration with 5 μg/dL intervals. Multiple linear regression analysis was used to adjust the influence factors age, sex, BMI, fasting glucose, smoking, and drinking habits on the inhibition of ALAD enzyme activity. Changes of ALAD enzyme activity were compared between groups 2–9 based on the lowest mean blood lead concentration (blood lead < 5, group 1). As shown by the results in Table 3, there was no significant difference in ALAD enzyme activity between groups 2 and 1. We found that when the concentration of blood lead in group 3 reached the range of 10.01 to 15.0 μg/dL, ALAD enzyme activity began to be significantly lower than group 1. In other words, when blood lead was less than 10 μg/dL, there was no significant inhibition of ALAD enzyme activity, which could represent a threshold level. When the blood lead concentration exceeded this threshold level, the inhibition of ALAD enzyme activity could be observed clearly.

## 4. Discussion

In 1996, Sakai et al. conducted univariate linear regression of ALAD enzyme activity and blood lead in 191 male workers with blood lead concentration ranging from 2.5 to 115.4 μg/dL. This result indicated a highly inverse correlation between ALAD enzyme activity and blood lead concentration [22]. Campagna et al. reported that a potential lead threshold was identified at about 4.8 μg/dL in cord blood, above which the mother’s ALAD may be inhibited [23]. Our study had similar results for the relationship between blood lead concentration and ALAD enzyme. According to the analyses by GAMs and multiple regressions, a blood lead level (BLL) threshold around 5 μg/dL was suggested, which is easily applicable in the management of occupational health in the lead industry.

The results of multiple regression with categorized BLL in our study (Table 3) showed that an average blood lead concentration exceeding 10 μg/dL would significantly inhibit ALAD enzyme activity. Nordman et al. also found that when blood lead concentration exceeded 10 μg/dL, ALAD enzyme activity showed a significant inverse correlation [24], which was similar to the results in our study. However, according to the study of Wada et al., when blood lead concentration reached about 15 μg/dL, ALAD enzyme activity was significantly inhibited by lead [25], which was slightly higher than the results of our study. This might be due to the different method used to measure ALAD enzyme activity in their research. In 2003, Murata et al. used a benchmark dose method to study 186 lead workers with blood lead concentration between 2.1 and 62.9 μg/dL, and found that in workers with low levels (less than 10 μg/dL), there was no significant effect on ALAD enzyme activity compared to higher blood lead levels [26]. In 2017, a Mexican study reported that 633 pregnant women aged 13 to 43 years showed even lower LAD activity if their blood lead level was between 2.2 and 10 ug/dL. One reason might be that pregnancy increases susceptibility to oxidative stress because of the mitochondria-rich placenta [27]. In our study, using GAMs with smoothing curves and multiple linear regression analyses of categorized blood lead levels to adjust for the potential confounding factors age, BMI, fasting blood glucose, and smoking and drinking habits, we found that there was no significant inhibition of ALAD enzyme activity with blood lead levels lower than 10 ug/dL. Thus, a blood lead level between 5 and 10 ug/dL would be an inflection point for declining ALAD activity among adults. We would suggest setting a cut-off value of 5 ug/dL BLL.

In the previous literature, Ratnaike et al. found that ALAD activity was dependent on the formation of Schiff base (a nitrogen analog of an aldehyde or ketone) between the substrate ALA and the amino group of a specific lysine residue of the enzyme. When this lysine residue was glycated by Schiff base formation with glucose, it led to a loss of ALAD activity, which showed that blood glucose inhibits the activity of ALAD enzymes [28]. After adjusting the other confounding factors, we also found similar results in our study: with an increase of 1 mg/dL fasting blood glucose, ALAD enzyme activity was reduced by 0.11 U/L (*p* < 0.001).

It was also noted in the previous literature that smoking and drinking alcohol directly affected ALAD enzyme activity [29], however, only drinking alcohol (twice per week) was found to have a significant marginal effect (*p* = 0.049) in our study. One reason might be that the measurement of smoking and drinking was not precise enough, which was limited by the design of our study, evaluating these factors by questionnaire. Another possible reason was that both lead and alcohol deplete vitamin B12 and folate in serum, which may concomitantly affect the ALAD activity [7]. In addition, increasing oxidative stress from lead exposure and smoking may also affect ALAD enzyme activity via the prooxidant–antioxidant disequilibrium [30].

According to the distribution of the ALAD G177C (rs1800435) polymorphism in the Taiwanese population as determined by Hsieh et al., there were 630 ALAD 1-1 phenotypes (95.4%), 29 ALAD 1-2 (4.4%), and only 1 ALAD 2-2 (0.2%) [10]. Yang et al. showed a consistent distribution of ALAD variants in Han workers in southwestern China (n = 156), for which the allele frequencies of ALAD 1 and ALAD 2 were 0.9679 and 0.0321, respectively [31]. Comparing the genotype distribution of ALAD G177C (rs1800435) between our study and Hsieh et al., the results showed no significant difference (*p* = 0.222) using the chi-square test. Due to the extremely low proportion of ALAD 2-2 genotypes in the Taiwanese population, we could not investigate the modified effect of ALAD G177C (rs1800435) polymorphism on blood lead levels and ALAD enzyme activity.

## 5. Conclusions

The results of our study show that ALAD enzyme activity had a significant dose-related relationship with blood lead; when the blood lead concentration exceeded 5 μg/dL, ALAD enzyme activity had a significant inhibitory effect, and this might represent a threshold value. The novelty of our finding would be the proposal that ALAD could be used as an indicator for lead exposure regulation.

## Figures and Tables

**Figure 1 ijerph-17-05712-f001:**
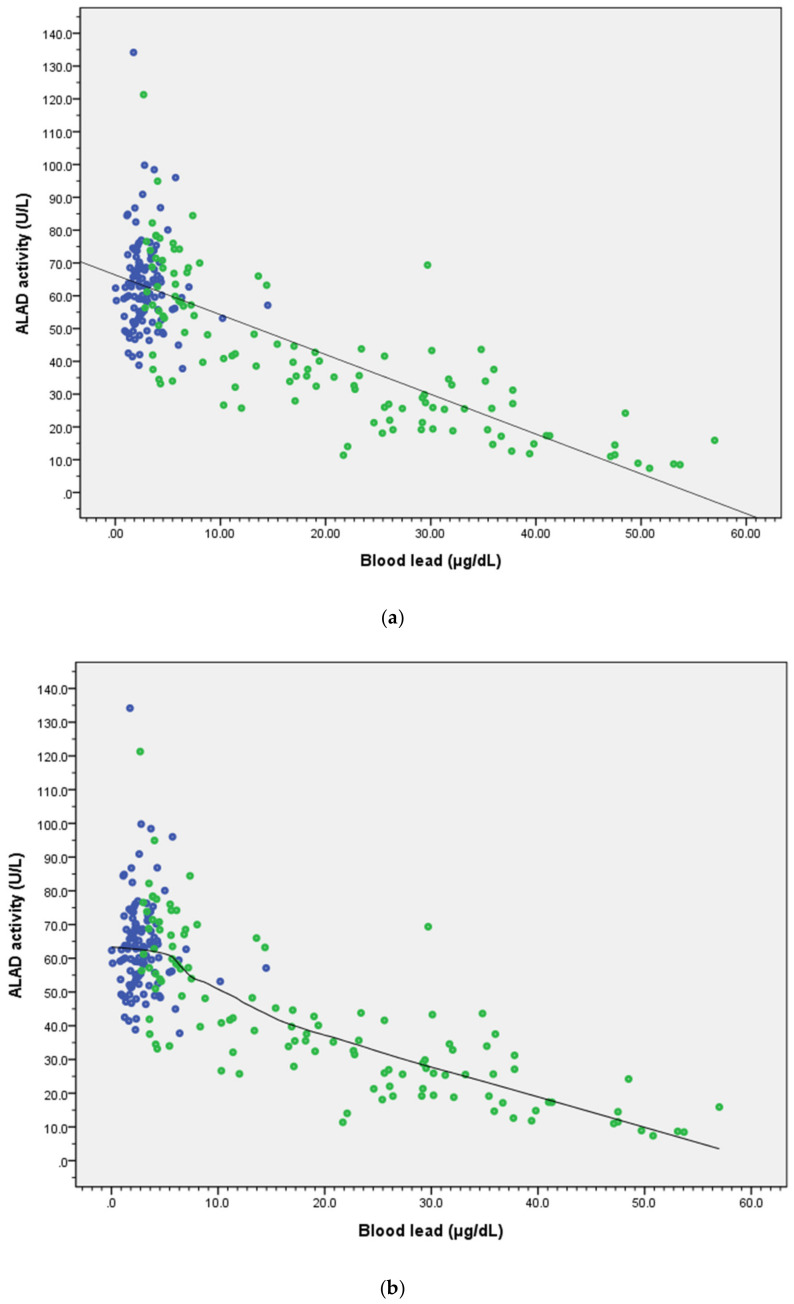
Association of blood lead concentration (ug/dL) and δ-aminolevulinic acid dehydratase (ALAD) activity in lead-exposed group (green dots, *n* = 116) and control group (blue dots, *n* = 113). (**a**) Multiple regression found a dose–response relationship. (**b**) Generalized additive models (GAMs) and scatter plots with smoothing curves (loess, span: 70%) used to look at the curve shape of the relationship between ALAD activity and blood lead level found an inflection point around 5 ug/dL. Both models were adjusted for gender, age, BMI, fasting blood sugar, tobacco smoking, and alcohol drinking (more than twice a week).

**Table 1 ijerph-17-05712-t001:** Comparison of the characteristics of the control and lead-exposed groups.

Characteristics	Control	Lead-Exposed	*p*-Value
	*n* = 117	*n* = 121	
Age (years)	41.7 ± 11.8	41.5 ± 8.3	0.916 ^a^
Time on the job (years)	–	11.7 ± 5.3	–
Gender (male %)	71 (61.1%)	98 (81.0%)	0.01 ^b^
BMI (kg/m^2^)	24.9 ± 4.1	24.4 ± 3.5	0.310 ^a^
Blood lead (μg/dL)	2.9 ± 1.8	19.7 ± 14.7	<0.001 ^a^
Hemoglobin (mg/dL)	14.7 ± 1.5	14.9 ± 1.4	0.168
Hematocrit (Hct) (%)	43.5 ± 3.8	44.4 ± 3.5	0.060 ^a^
ALAD activity (U/L)	63.32 ± 14.01	41.58 ± 22.07	<0.001 ^a^
ALAD 1-1 type (%)	113 (96.6%)	116 (95.9%)	0.773 ^b^
ALAD 1-2 type (%) ^†^	4 (3.4%)	5 (4.1%)	
Fasting glucose (mg/dL)	106.7 ± 38.6	101.8 ± 32.3	0.309 ^a^
Alcohol use (%)	9 (7.7%)	32 (26.4%)	<0.001 ^b^
Smoking (%)	16 (13.7%)	49 (40.5%)	<0.001 ^b^

^a^ T-test; ^b^ chi-square test; **^†^** no subjects of ALAD 2-2 type. BMI—body mass index; ALAD—δ-aminolevulinic acid dehydratase.

**Table 2 ijerph-17-05712-t002:** Multiple linear regression model for ALAD activity.

	All (*n* = 238)			ALAD 1-1 Type (*n* = 229)		
Variables	Regression Coefficient	(SE)	*p*-Value	Regression Coefficient	(SE)	*p*-Value
Blood lead (μg/dL)	−1.04	(0.08)	<0.001 **	−1.05	(0.08)	<0.001 **
ALAD type (1-2 vs. 1-1)	−4.12	(4.75)	0.39	–	–	
Gender (M vs. F)	−8.08	(2.27)	<0.001 **	−8.28	(2.31)	<0.001 **
Age (years)	−0.16	(0.09)	0.08	−0.19	(0.09)	0.05
BMI (kg/m^2^)	0.20	(0.26)	0.45	0.33	(0.27)	0.23
Fasting glucose (mg/dL)	−0.11	(0.03)	<0.001 **	−0.12	(0.03)	<0.001 **
Smoking (yes/no)	0.96	(2.37)	0.68	1.22	(2.40)	0.61
Drinking (yes/no)	−5.47	(2.69)	0.04 *	−5.51	(2.70)	0.04 *
Constant	83.84	(6.76)	<0.001 **	82.86	(6.91)	<0.001 **
	(adj R^2^ = 0.639)			(adj R^2^ = 0.644)		

* *p* < 0.05; ** < 0.01

**Table 3 ijerph-17-05712-t003:** Multiple linear regression analyses for association with ALAD activity and blood lead level (ALAD 1-1 type only, n = 229). Adjusted for gender, age, BMI, fasting glucose, smoking, and drinking habits.

Group	Range of Blood Lead (μg/dL)	No. of Subjects	ALAD Activity (U/L)	Blood Lead (μg/dL)	β	SE	*p*-Value
1	≦5.00	128	64.21 ± 15.04	2.74 ± 1.11	–	–	–
2	5.01–10.00	26	60.49 ± 14.15	6.48 ± 0.93	−1.099	2.925	0.708
3	10.01–15.00	12	44.64 ± 13.30	12.15 ± 1.60	−16.328	4.004	<0.001
4	15.01–20.00	11	37.76 ± 5.36	17.65 ± 1.24	−23.710	4.237	<0.001
5	20.01–25.00	8	28.18 ± 11.40	22.66 ± 1.15	−33.514	5.041	<0.001
6	25.01–30.00	13	28.89 ± 13.65	27.58 ± 1.77	−31.159	3.996	<0.001
7	30.01–35.00	9	29.92 ± 9.27	31.73 ± 1.55	−34.900	4.569	<0.001
8	35.01–40.00	11	22.33 ± 9.14	37.05 ± 1.58	−38.096	4.271	<0.001
9	40.01	11	13.20 ± 5.17	48.84 ± 4.88	−45.991	4.594	<0.001

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
