# Peer review of "Use of Generalized Additive Model to Detect the Threshold of δ-Aminolevulinic Acid Dehydratase Activity Reduced by Lead Exposure"

_ijerph, 2020, doi:10.3390/ijerph17165712_

Round 1

Reviewer 1 Report

Thank you for 

Using smoothing methods to identify a threshold for dangerous blood lead content is a very good idea. Using a control group to have more distribution in the exposure is a good idea.However, the aims are not clear, is there anything in the lead exposure literature that leads you to think that there is a threshold? Please find a more relevant reference or explain your rationale for referring to Chuang et al. (about vibrations).

Methods

There should be information on adjustment of the regression analysis presented in figure 1ab. As the proposed threshold effect seems to occur at 5µg/dL Lead, the threshold could be due to differences in the control and study population, so it would be desirable to see results that are adjusted for lifestyle factors mentioned in table 1, or a comparison between unadjusted and adjusted results. 

Results

In Figure 1 a and b, please denote who are controls and who are the exposed group. Also, you could plot the lines on the same graph and save space. And present a fit statistic (R2, AIC?) of the different lines for comparison.

It is ambitious to be showing genotype-specific effects when the genotype is very rare, and the results reflect this as the study is underpowered to show any specific effects, the results in table 2 could be written out.

Discussion/Conclusion

Concepts such Schiff base formation should be defined/explained in a general environmental health journal such as IJERPH.

Blood glucose is not mentioned in the aims, so it is surprising to see it mentioned as an important conclusion.

The references are very old and a simple literature search found other articles which deal with threshold values, albeit in pregnant women.

https://pubmed.ncbi.nlm.nih.gov/10403633/

Reviewer 2 Report

Line 35 – The authors should revise the sentence were is written “decreasing hemoglobin or anemia”, since decreasing hemoglobin can lead to anemia; Lead can also cause iron deficiency resulting also in anemia, and this is only one of the several mechanisms of lead induced anemia; In the same perspective, attention should be addressed in line 48 concerning the sentence “determination of ALAD activity was well suited as a measure of lead-induced anemia”.

Line 67 – Are there published results regarding susceptibility to Pb of the Taiwan ALAD studied genotypes? If so, the authors should mention them.

Lines 166 and 167 – sd values should also be mentioned in the text.

Line 168- review the sentence.

Line 178 – where is written “drinking twice per-week”, I believe its alcohol?

Table 2 – Flagging all p-values<0.05 would turn the interpretation of the table easier for the reader (e.g. using *).

Line 251 – If available, a brief explanation of the mechanisms through which ALAD is inhibited by tobacco (I believe the presence of Pb) and alchool, would improve the paper.

Line 267 – When the authors write in the conclusions that “The results of our study showed that ALAD enzyme activity is highly sensitive to lead inhibition”, this is already known. Considering that it is in the topic “Conclusions”, the sentence should be revised. The novelties of this work are the proposal that ALAD which may be used as an indicator of lead exposure regulation and that “when the blood lead concentration exceeds 5 μg / dL, ALAD enzyme activity would have a significant inhibitory effect, and might be a threshold of limit value”. This is what must be highlighted in conclusions.

Reviewer 3 Report

Thanks for inviting me to review the manuscript intitled “Use of Generalized Additive Model to Detect the Threshold of δ-Aminolevulinic Acid Dehydratase Activity Reduced by Lead Exposure.” Huang et al have proposed that ALAD enzyme activity is overly sensitive to lead inhibition and has a significant dose-related relationship with blood lead, which may be used as an indicator of lead exposure regulation. My suggestions are as follow:

First, Language should be better adapted to scientific writing.

Second, L164-166: Authors should provide the standard deviation of blood lead concentrations and ALAD activity.

Third, figure quality should be enhanced and better organized.

Fourth, have the authors investigated the impact of years in the job, and blood lead levels and/or ALAD activity?, Because time of exposure to Pb is associated with blood lead levels increased:

Himani, Kumar, R., Ansari, J.A. et al. Blood Lead Levels in Occupationally Exposed Workers Involved in Battery Factories of Delhi-NCR Region: Effect on Vitamin D and Calcium Metabolism. Ind J Clin Biochem 35, 80–87 (2020). https://doi.org/10.1007/s12291-018-0797-z

Although authors did not investigate the perspectives on ALAD 1-2 workers, I suggest a previous study that demonstrated an association between ALAD 1-2 carriers and high blood lead levels, and also reported high ALAD activity in ALAD-1-1 carriers, which should be addressed to the discussion of the authors:

Mani, M. S., Kunnathully, V., Rao, C., Kabekkodu, S. P., Joshi, M. B., & D’Souza, H. S. (2018). Modifying effects of δ -Aminolevulinate dehydratase polymorphism on blood lead levels and ALAD activity. Toxicology Letters, 295, 351–356. doi:10.1016/j.toxlet.2018.07.014 
